# A Putative TRAPα Protein of Microsporidia *Nosema bombycis* Exhibits Non-Canonical Alternative Polyadenylation in Transcripts

**DOI:** 10.3390/jof9040407

**Published:** 2023-03-25

**Authors:** Yujiao Wu, Ying Yu, Quan Sun, Yixiang Yu, Jie Chen, Tian Li, Xianzhi Meng, Guoqing Pan, Zeyang Zhou

**Affiliations:** 1State Key Laboratory of Resource Insects, Southwest University, Beibei, Chongqing 400715, China; 2Chongqing Key Laboratory of Microsporidia Infection and Control, Southwest University, Beibei, Chongqing 400715, China; 3Key Laboratory of Sericultural Biology and Genetic Breeding, Ministry of Agriculture and Rural Affairs, Southwest University, Chongqing 400715, China; 4Key Laboratory of Conservation and Utilization of Pollinator Insect of the Upper Reaches of the Yangtze River (Co-Construction by Ministry and Province), Ministry of Agriculture and Rural Affairs, Chongqing Normal University, Chongqing 400047, China

**Keywords:** microsporidia, *Nosema bombycis*, TRAPα, alternative polyadenylation

## Abstract

Microsporidia are obligate intracellular eukaryotic parasites that have significantly reduced genomes and that have lost most of their introns. In the current study, we characterized a gene in microsporidia *Nosema bombycis*, annotated as *TRAPα* (*HNbTRAPα*). The homologous of TRAPα are a functional component of ER translocon and facilitates the initiation of protein translocation in a substrate-specific manner, which is conserved in animals but absent from most fungi. The coding sequence of *HNbTRAPα* consists of 2226 nucleotides, longer than the majority of homologs in microsporidia. A 3′ RACE analysis indicated that there were two mRNA isoforms resulting from non-canonical alternative polyadenylation (APA), and the polyadenylate tail was synthesized after the C^951^ or C^1167^ nucleotide, respectively. Indirect immunofluorescence analysis showed two different localization characteristics of *HNbTRAPα,* which are mainly located around the nuclear throughout the proliferation stage and co-localized with the nuclear in mature spores. This study demonstrated that the post-transcriptional regulation mechanism exists in Microsporidia and expands the mRNA isoform repertoire.

## 1. Introduction

Microsporidia is a phylum with diverse single-celled intracellular parasites infecting almost all the animals and some protists [1,2,3,4,5]. They are characterized by unique infection apparatus, which consist of a polaroplast, polar tube, and posterior vacuole, as well as a thick chitin layer spore wall. The genomes of microsporidia range from 2.3 Mbp to 51.3 Mbp, and the microsporidium *Encephalitozoon intestinalis* has the smallest known eukaryotic genome [3,6,7,8]. Microsporidia have lost many genes related to metabolic and regulatory pathways, and even have drastic reductions in the length of non-coding regions (i.e., intergenes and introns) [6]. Many microsporidia have tiny intergenic regions (IGRs), and the IGRs of the *E. intestinalis*, for example, have an average length of only 115 base pairs [9], with coding regions accounting for 90% of the entire genome [10]. The length of the IGRs can even be negative, and the genes can overlap with each other [3,9,11]. Some genera of microsporidian have lost both splicing machinery and introns [12,13], while others that retain introns have reductions in both their number and length [14,15]. Microsporidia prefer to retain secreted proteins like hexokinase and trehalase [16]. These secreted proteins enter the host’s metabolic pathway and cause upregulation of the nucleotide, amino acid, and fatty acid biosynthesis in the host [17,18]. Plentiful transporters, which are located at the interface between the host and the parasite, allow the microsporidia to steal ATP and metabolites from the host [6,19]. Microsporidian genome evolution is a highly dynamic process that has balanced constraint, reductive evolution, and genome expansion during the adaptation to an extraordinarily successful obligate intracellular lifestyle [20]. Microsporidia are fascinating in that they produce unique spores that are one of the most complex single-celled forms known to the biological world [21]. Due to the unique structure of spores, which are equipped with an infection apparatus and are remarkable in a wide range of hosts, detailed research on microsporidia will deepen our understanding of pathogens.

*Nosema bombycis* is a well-known microsporidium and the major pathogen of the silkworm *Bombyx mori*, which once caused catastrophic damage in sericulture. Herein, we found a protein annotated as the translocon-associated protein TRAPα in *N. bombycis* and demonstrated the existence of a non-canonical alternative polyadenylation (non-APA) in its transcription process. Alternative polyadenylation (APA) is a widespread mechanism used to control gene expression. APA occurs when a single gene has multiple poly(A) sites, and regulates the function, stability, localization, and translation efficiency of the genes by generating various isoforms of transcripts [22]. Depending on where the different poly(A) sites are located in the genes, APA events are separated into two major categories: the untranslated region APA and the coding region APA [23]. Generally, polyadenylation events occur at the 3′ untranslated region (3′ UTR), but APA may also occur at different regions inside the loci (5′ UTRs, introns or exons), which are known as non-canonical sites [24]. Extracellular stimulation affects the selection of poly(A) sites, by influencing the expression of proteins involved in polyadenylation [25]. The APA closer to the 5’ end, indicates that the mRNAs have shorter 3′ UTRs, which means that the mRNAs get longer half-lives and produce more proteins. This approach can help cells respond to changes in the external environment [26].

The translocon-associated protein (TRAP), originally identified as a signal-sequence receptor (SSR), is a tetrameric complex substrate that specifically facilitates the initiation of protein translocation [27,28,29,30,31]. TRAPα is a single-spanning membrane glycoprotein of molecular weight ~34 kDa and displays a remarkable charge distribution, with the *N* terminus highly negatively charged and *C* terminus positively charged [32]. TRAPα can be cross-linked to nascent chains and membrane-bound ribosomes [30,33]. Hirama et al. first reported the TRAPα transcripts induced by granulocyte-macrophage colony-stimulating factor in the growth factor-dependent human myeloid cell line TF-1 and exhibited a complicated non-canonical alternative polyadenylation (APA) [34].

In this study, the hypothetical translocon-associated protein HNbTRAPα in *N. bombycis* exhibited non-canonical alternative polyadenylation, which showed two different localization patterns at different developing stages of *N. bombycis*. The analysis will enhance our understanding of the adaptive evolution and post-transcriptional regulation of the microsporidian genome.

## 2. Materials and Methods

### 2.1. Spores Purification

The fifth instar silkworm larvae infected with 10^5^ spores of *N. bombycis* were collected at the pupal stage. Pupas discarded epidermis and intestines were homogenized and diluted with the appropriate amount of sterilized ddH_2_O and, subsequently, filtered with gauze, followed by cotton. The filter liquor was centrifuged at 500 rpm for 5 min at room temperature, then the supernatant was transferred to a new tube and recentrifuged at 3000 rpm for 5 min at room temperature to collecting precipitation. After 3 instances of differential centrifugation, the sediments were resuspended with sterilized ddH_2_O. The primary extracted spores were filtered with new cotton twice and resuspended in the appropriate amount of sterilized ddH_2_O. The spore suspension was delivered to the Percoll discontinuous density gradient centrifugation (30%, 45%, 60%, 90%, *v*/*v*) and centrifuged at 16,000× *g* at 4 °C for 40 min. Separated spores in a 90% layer (mainly mature spores) were collected and washed with sterile ddH_2_O no less than 3 times, until no Percoll remained. The purified spores were stored as precipitation in just enough sterilized ddH_2_O with penicillin-streptomycin antibiotics (100 U/mL) at 4 °C to maintain their vitality.

### 2.2. Genomic DNA and Total RNA Extraction

The fourth instar silkworm larvae were infected with 10^4^ spores of *N. bombycis* per individual by oral administration and reared to the fifth instar stage. The midgut at the late stage of the fifth instar larvae were collected and frozen immediately with liquid nitrogen. A portion of the midgut was ground and the infection by *N. bombycis* was detected by microscopy. Then, three severely infected tissues were ground in liquid nitrogen. A moderate powdered mixture was immediately transferred into an RNase-free centrifuge tube, which had CPL/β-mercaptoethanol added beforehand. The genomic DNA and total RNA of the infected midgut were extracted using the Omega EZNA^TM^ Plant DNA/RNA Kit (R6733-01), following the instructions. 

The genomic DNA of the purified *N. bombycis* spores was extracted using the CTAB method. Briefly, 10^8^ of *N. bombycis* spores were incubated with 2% CTAB buffer (8.0 g cetyltrimethylammonium bromide together with 32.728 g NaCl, 40 mL 1M Tris-HCl (pH8.0), and 16 mL 1 M EDTA were dissolved in ddH_2_O to a final volume of 400 mL) and protease K at 65 °C for 2 h. After being centrifuged at 6000 rpm for 1 min, the supernatant was transferred to a new tube for routine phenol chloroform extraction. The final purified genomic DNA was dissolved in sterilized ddH_2_O and stored at −20 °C.

### 2.3. 3′ RACE Analysis

The 3′ terminal of *HNbTRAPα* was analyzed using the 3′ Full RACE Core Set with the PrimeScript™ RTase kit (TaKaRa, Code No. RR002A, Maebashi, Japan). Four specific primers were designed according to the sequence information, each of which was named the following: *HNbTRAPα*-GSP1, *HNbTRAPα*-GSP2, *HNbTRAPα*-GSP3, and *HNbTRAPα*-GSP4 (Table 1). These primers were combined in pairs, as GSP1 (outer primer)-GSP2 (inner primer), GSP1 (outer primer)-GSP3 (inner primer), GSP1 (outer primer)-GSP4 (inner primer), GSP2 (outer primer)-GSP3 (inner primer), GSP2 (outer primer)-GSP4 (inner primer), and GSP3 (outer primer)-GSP4 (inner primer), respectively, for the nested PCR. Using the *N. bombycis* total RNA extracted from infected silkworm midguts as the template, the 1st strand cDNA was synthesized by using the 3′ RACE adaptor primer for reverse transcription. The total RNA used in each reverse transcription reaction was 1 μg. The 1st PCR reaction was performed using GSP1~GSP4 as the specific upstream outer primers and the 3′ RACE outer primer as the downstream primer. If the target product was not obtained through the 1st PCR reaction, the 2nd PCR reaction was performed using GSP2~GSP4 as the specific upstream inner primers and the 3′ RACE inner primer as the downstream primer. The detail of the operation was referred to in the specification of the kit. The PCR products were cloned into the pBluescript II KS (-) vector and sequenced later.

### 2.4. Pathogen Inoculation

The fresh silkworm pupa infected by *N. bombycis* was wiped with 75% alcohol twice, then soaked in 3% H_2_O_2_ solution for 1 min. Then, the pupa was immediately transferred into enough sterilized ddH_2_O, at least three times, avoiding the damage to spores caused by H_2_O_2_. The hemolymph, which was filled with *N. bombycis* at different development phases, was collected at the somite of the pupa through puncturing with a sterilized micropipette. The hemolymph was mixed with the right amount of *N*-Phenylthiourea and centrifuged at 3000 rpm for 2 min. Afterwards, the precipitates containing the spores were washed with sterilized 1× PBS several times and resuspended with the Grace medium (Gibco, Billings, MT, USA). The counted spores were added to the Sf9 or BmE-SWU cells in a 10:1 ratio. Infected cells were cultured at 28 °C in the Grace medium with 10% fetal bovine serum (Gibco, Billings, MT, USA).

### 2.5. The Specific HNbTRAPα-Antisera Preparation

Specific primers (Table 1) were designed using the Primer Premier 5 software [35], and the forward and reverse primers were added with the BamHI and Sal I restrictions sites individually. Then, the fragments of the *N* terminus (1–696 bp), central region (697–981 bp), and *C* terminus (1704–2226 bp) of the HNbTRAPα were amplified, using the *N. bombycis* genomic DNA as a template. The PCR products purified by agarose gel recovery were cloned into the pCold I vector, which was digested by BamHI and Sal I. Recombinant plasmids were transformed into Escherichia coli Rosetta competent cells. The single colony containing the pCold I-HNbTRAPα plasmid was expanded in an LB medium. Then, the recombinant bacteria at the log-growth stage was induced to express recombinant HNbTRAPα (rHNbTRAPα) by adding 0.1 mM IPTG (isopropyl-bd-thiogalactopyranoside) and cultured at 16 °C for 24 h. The collected bacteria were broken up with ultrasonic vibration and the precipitation was removed by centrifugation. The supernatant containing rHNbTRAPα was purified using Ni Sepharose (GE, New York, NY, USA). The concentration of the final obtained rHNbTRAPα was detected using the Bradford method. Then, 100 μg of recombinant protein dissolved in PBS (pH 7.4) was emulsified with an equal volume of Freund’s adjuvant (Sigma-Aldrich, Saint Louis, MO, USA). Balb/c mice were immunized subcutaneously every week. Complete Freund’s adjuvant was used at the first immunization and incomplete Freund’s adjuvant was used the remainder 3 times. Then, antisera were obtained one week after the fourth immunization and stored at −20 °C. The antibody titer was detected using the indirect ELISA.

### 2.6. Immunoblottings

Proteins extracted from the mature spores and *N. bombycis* infected Sf9 cells were subject to 12% SDS-PAGE. The separated proteins were transferred on a PVDF membrane and used for immunoblotting. After blocking, an anti-HNbTRAPα-C serum with a dilution of 1:500 and a HRP-conjugated goat anti-mouse IgG (Sigma, Saint Louis, MO, USA) with a dilution of 1:8000 in a blocking buffer were incubated with a membrane for 0.5 h at room temperature, separately. The membrane, which was washed three times, was detected with Clarity Western ECL Substrate (Bio-Rad, Hercules, CA, USA).

### 2.7. Immunofluorescence Analysis

The cells Infected after 3, 6, 12, and 17 days were collected, respectively, and cultured on coverslips at 28 °C for 2 h. The cells were washed twice in 1× PBS (pH 7.4) and fixed with 4% formaldehyde at room temperature for 10 min. After PBS washing, the fixed cells were handled before permeabilization in PBS-1% TritonX-100 for 20 min. Then the samples were washed three times in PBST (PBS + 0.5% Tween 20) for 5 min each time. The cells were incubated with PBST containing 10% goat serum and 5% BSA as a blocking buffer at room temperature for 1 h. After being washed three times, as above, the samples were co-incubated with mouse anti-HNbTRAPα serum and rabbit anti-Nbtubulin-*β* serum [36], with a dilution rate of 1:100 in blocking buffer, for 1 h at room temperature. The normal serum of the mouse and rabbit were used as negative controls. Followed by washing three times, the samples were co-incubated with Alexa Fluor^®^ 488 conjugate goat anti-mouse IgG and Alexa Fluor^®^ 594 conjugate goat anti-rabbit IgG (Thermo Fisher, Waltham, MA, USA), as well as DAPI (4′6-diamidino-2-phenylindole, Sigma, 1:1000 diluted) for 40 min at room temperature. After being washed three times, the samples with added ProLong^®^ Gold antifade reagents (Thermo Fisher, Waltham, MA, USA) were sealed and imaging was carried out using an Olympus FV1200 laser scanning confocal microscope.

## 3. Results

### 3.1. Sequence Analysis of HNbTRAPα

The open reading frames of *HNbTRAPα* (Genbank. OQ632563) are 2226 bp in length and, theoretically, they encode a protein with a molecular weight of 86 kDa and pI of 8.78. The sequence of *HNbTRAPα* is nearly twice as long as other genes, except the ones of *Vittaforma corneae*. Although multiple sequence alignment showed HNbTRAPα had low similarity with homologs in other microsporidia and TRAPα in fungi, the N terminal sequences of microsporidian HTRAPα are comparatively conserved. TRAPα was characterized by a negatively charged *N* terminal in mammalian and *Oncorhynchus mykiss*, which played a regulatory role through combining with the calcium existing in ER lumen. However, the number of negative residues in HNbTRAPα was not dominant (Appendix A). The phylogenetic tree further indicated that the evolution of HNbTRAPα was independent of fungi and protozoa, and not conserved with reported mammalian sequences. HNbTRAPα in microsporidia was clustered into a solo branch. Functional domain prediction indicates the existence of the nucleus localization motif in microsporidia and mammalian hosts. It is remarkable that there was no transmembrane helix and signal peptides predicted in HNbTRAPα, while they were relatively conserved among others of the mammalian and other species (Figure 1).

### 3.2. Two Transcriptional Isoform of HNbTRAPα

Considering the sequence length of *HNbTRAPα* is much longer than other genes, and there are three coding sequences with different lengths in the genome database of *N. bombycis*, we performed the 3′ RACE to analyze whether alternative polyadenylation existed in this gene. After the outer PCR reaction using the 5′ specific primer GSP1, GSP2, GSP3, and GSP4, we obtained two weak bands amplified by the primer GSP2/3′ RACE outer primer and the GSP4/3′ RACE outer primer, respectively. Hence, we processed the inner PCR reaction using those four outer PCR products as templates and obtained two strongly specific *HNbTRAPα* sequences, as detected on agarose gel (Figure 2). The fragments were recovered and purified, and successfully cloned on the pBluescript II KS (-) plasmid for sequencing analysis. Finally, six sequences containing specific primer GSPs were obtained, but only four sequences could be matched to *HNbTRAPα*. As the multiple alignment result showed, there are two types of polyadenylated mRNA existing in *HNbTRAPα* transcripts, and the two identified polyadenylation sites are C^951^ and C^1167^, respectively (Appendix A). Further, sequence analysis showed that there was no canonical polyadenylation signal (AAUAAA) in this sequence, suggesting a non-canonical alternative polyadenylation existed in this gene.

### 3.3. Three Protein Productions of HNbTRAPα Were Detected in N. bombycis

Antiserum against HNbTRAPα was used to display the translation products of the gene. As shown in the Western blot results, the antiserum against peptides 569–742 (C ter-minal, HNbTRAPα-C) can mainly recognize a ~37 kDa protein band and a weakened ~86 kDa in purified mature spores (Figure 3A, lane 2). Meanwhile, in infected Sf9 cells containing *N. bombycis* that are predominant in the proliferative phase, HNbTRAPα-C antiserum can specifically bind with a ~45 kDa protein (Figure 3A, lane 3, and Figure 3B). The protein ~37 kDa, ~45 kDa and ~86 kDa are consistent with the predicted weight of the isoform that polyadenylated at 951 bp, 1167 bp, and the full length of HNbTRAPα.

### 3.4. HNbTRAPα Localized at the Perinuclear Region or in the Nucleus during Different Development Stages

To test whether HNbTRAPα could transport into the nucleus as a bioinformation prediction, we detected the location characteristic of HNbTRAPα at different development phases through an indirect immunofluorescent assay. The polyclonal antisera of the *N* terminal (1–232 aa), central region (233–327 aa), and C terminal (569–742 aa) of HNbTRAPα were prepared. The antiserum against Nbtubulin-*β* of *N. bombycis* was used to indicate the cytoplasm of meront [36]. Different development stages of *N. bombycis* were easily obtained when they were cultured in Sf9 or BmE cell lines. As the results showed, when the parasite was in the proliferative phase with a loose and big nucleus, the fluorescence signals of HNbTRAPα were distributed partially or wholly in the perinuclear region without or partially co-localized with β-tubulin. While in mature spores, the signal of HNbTRAPα totally co-localized with the nucleus (Figure 4 and Figure 5).

## 4. Discussion

Facing the complex environment in the cell, microsporidia suffer great selection pressure, which drives the evolution of microsporidia. The microsporidian genome extremely reduced the strategy of losing or shortening genes and the discarding of non-coding material, like introns and intergenic regions [37]. There was even no evidence of spliceosomal machinery in some microsporidia, such as *Nematocida* spp. infecting a nematode [12,38], and *Edhazardia aedis* infecting a mosquito. While the genome of some species like *Nosema bombycis* expanded due to the proliferation of host-derived transposable elements, tandem and segmental duplication, as well as horizontally transferred genes [8]. Proteins predicted in these microsporidia are much more than in the small-genome species. Alternative polyadenylation (APA) is widespread in eukaryotes and is a majority mechanism for modulating gene expression, including protein diversification [39]. First, we identified that APA exist in the microsporidium *N. bombycis*. The mechanism may contribute to increasing protein abundance in response to the environment.

The translocon-associated protein complex TRAP anchor to the ER membrane and mediate the nascent peptide to cross the ER membrane for post-translational processing. The nuclear membrane localization characteristic of HNbTRAPα is a newly interesting finding. Bioinformatic analysis suggests that several well-known microsporidian TRAPα have one or more nuclear localization signals, but there have been no associated reports even for the widely studied species of humans and dogs. In our research, we found that there was non-canonical coding-region alternative polyadenylation in *HNbTRAPα* post-transcriptional processing, and then we observed HNbTRAPα distributed in the perinuclear region or in cytoplasm during the merogony phases and gathered in the nucleus when spores matured. During the development stages, HNbTRAPα was distributed around the nucleus or nuclear membrane of *N. bombycis* meronts, suggesting a potential function associated with protein synthesis or processing. When the spores tend to be mature, HNbTRAPα might transport into the nucleus and participate in spore dormancy. However, we cannot determine the localization characteristics and functions of each isoform at present.

Previous studies found that the translocon-associated protein complex TRAP appears to be conserved in animals but is not uniform throughout the eukaryotic kingdom. Plants and algae have a simplified subunit composition that lacks TRAPγ and TRAPδ, while in most fungi TRAP is completely absent [27]. *HNbTRAPα*, which we studied in *Nosema bombycis*, was predicted to produce a protein with a molecular weight close to that of the TRAP complex (90 kDa) in mammals. It is interesting that HNbTRAPα and its homologous proteins were retained in the evolution of microsporidia, while most genes necessary for primary metabolism were lost. Even though it can be speculated that TRAPα as a conserved protein is important for *N. bombycis*, the retaining of *TRAPα* and its close association to nucleus in space also seems puzzling as this gene is not necessary for many species.

As is already known, TRAPα is an ER membrane protein with its *N* terminus in the lumen of ER and its *C* terminus in the cytoplasm, because of the existence of the transmembrane helix [32]. Both HNbTRAPα and its homologous proteins in microsporidia were predicted to have no signal peptides and transmembrane domains. Despite the specificity of the microsporidium genome, and the adaptive evolution may also result in the loss of signal peptides and transmembrane domains; it is hard to say if the protein is TRAPα. In addition, conserved domain prediction showed a HUN (HPC2 and ubinuclein) domain (E-value: 2.66 × 10^−7^) in HNbTRAPα, that may belong to a histone tail-binding chaperone. Different histone chaperones have different functions, including histone storage, histone transport to the nucleus, and deposition of nucleosomes in conjunction with specific chromatin remodeling complexes [40,41,42]. The functions of HNbTRAPα need to be uncovered in the future.

In this study, we have demonstrated the non-canonical APA mechanism existence in the *HTRAPα* of microsporidium *N. bombycis*, which might lead to various localization characteristic. It has important significance to enrich our knowledge of the gene regulation of *N. bombycis* and may be applied to other fungus-like eukaryotes. It also may help us to explore the molecular evolution events involving *N. bombycis*.

## Figures and Tables

**Figure 1 jof-09-00407-f001:**
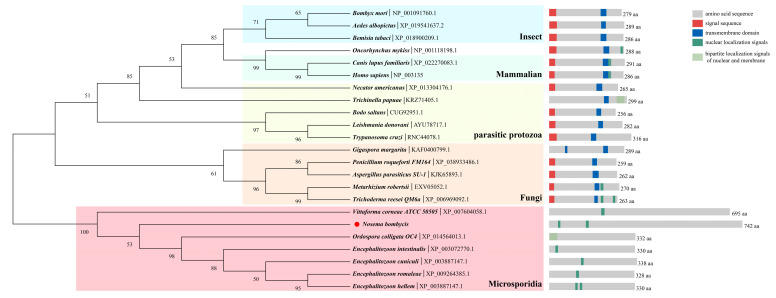
Phylogenetic analysis based on TRAPα in species and microsporidian HTRAPα using the maximum-likelihood method. Amino acid sequences downloaded from the NCBI database were used to construct a phylogenetic tree using MEGA 11 software. The Jones–Taylor–Thornton (JTT) model and bootstrap method with 1000 replications were selected to construct a maximum-likelihood tree. Signal peptide, transmembrane domain and nuclear localization signals were predicted by SignalP 5.0 (SignalP 5.0 Services, DTU Health Tech), DeepTMHMM (DTU/DeepTMHMM, BioLib), and cNLS Mapper (NLS Mapper (keio.ac.jp)), respectively. As the result shows, TRAPα from different species mainly clustered into four clades, and the HNbTRAPα clustered with other microsporidia homologues independently. The insects TRAPα are relatively close to mammalian ones. All microsporidia TRAPα lacked signal sequence and transmembrane domains. The amino acid sequence lengths and function domain sites shown in the diagram are drawn to scale. Each species name is followed by the GeneBank ID or NCBI accession number of the sequence.

**Figure 2 jof-09-00407-f002:**
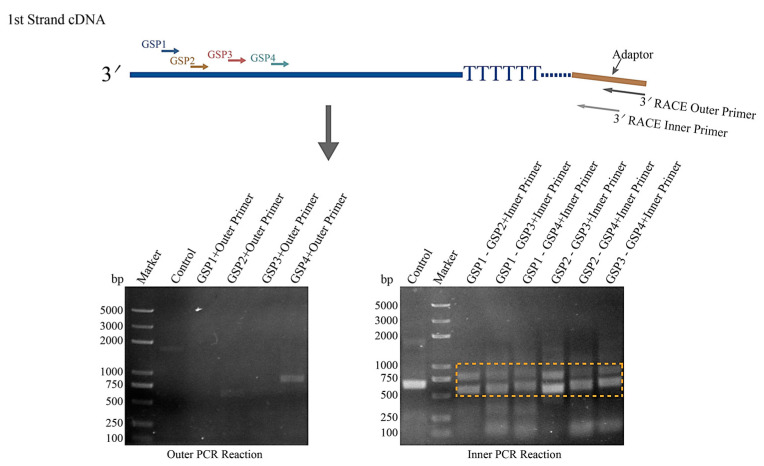
3′ RACE analysis of the transcription isoforms of *HNbTRAPα*. The nested PCR process of 3′ RACE analysis is shown in the schematic diagram. The 1st strand cDNA with an adaptor at the 3′ terminal was obtained by reverse transcription. The outer PCR reactions were performed using the 3′ RACE outer primer paired with GSP1, GSP2, GSP3, and GSP4, respectively. Then, the inner PCR reactions were performed using the outer PCR products as templates, and amplified with GSP2, GSP3, and GSP4 as specific upstream primers, respectively. The outer and inner PCR products were separated using 1.5% agarose gel. In the first amplification, only GSP2 and GSP4 obtained two weak bands at 500~750 bp and 750~1000 bp, respectively. Two strong bands (500~750 bp and 750~1000 bp) were detected after the inner PCR reaction, and the products indicated in the yellow rectangle were recovered and cloned, respectively, for sequencing.

**Figure 3 jof-09-00407-f003:**
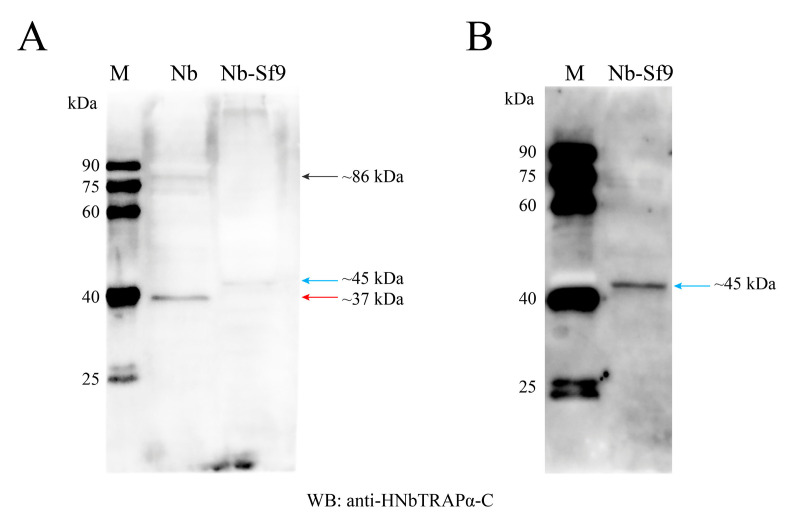
Western blotting assay of HNbTRAPα in *N. bombycis* mature spores and infected Sf9 cells that predominantly with the proliferative phase of *N. bombycis*. (**A**): The protein isoform detection of HNbTRAPα in mature spores and proliferative stage with anti-HNbTRAPα-C serum. The antiserum against HNbTRAPα-C can bind strongly with a ~37 kDa band and weakly with a ~86 kDa band in purified mature spores. While, this antiserum specifically recognized a ~45 kDa protein in Sf9 cells with *N. bombycis* infection. (**B**): The antiserum against HNbTRAPα-C strongly bound with the ~45 kDa protein and two weak protein bands in infected Sf9 cells under overexposure conditions. The protein ~37 kDa, ~45 kDa, and ~86 kDa are consistent with the predicted weight of the isoform that polyadenylated at 951 bp, 1167 bp and the full length of HNbTRAPα. M: Marker; Nb: Nb mature spores; Nb-Sf9: Nb infected Sf9 cells.

**Figure 4 jof-09-00407-f004:**
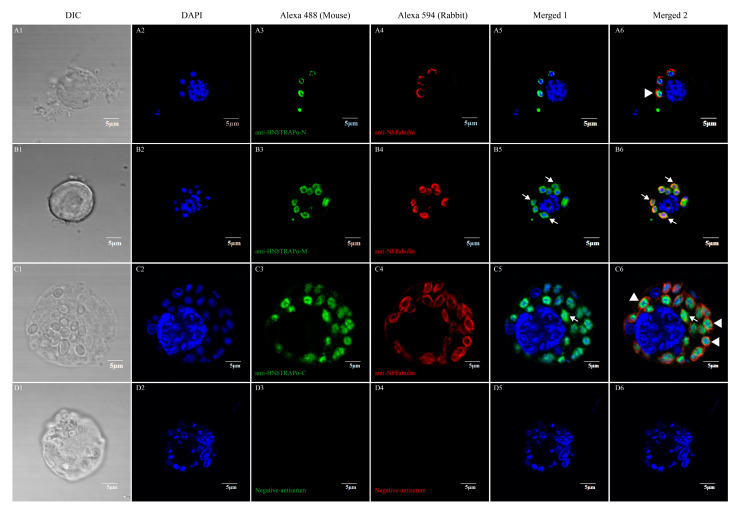
Indirect immunofluorescent assay of HNbTRAPα in developing stages within Sf9 cells. In developing stages, HNbTRAPα locates around or in the nucleus. A1~A6 showed the nuclear membrane distributed HNbTRAPα, which was detected by mouse antiserum against the N terminal (1–232 aa) of HNbTRAPα. B1~B6 showed the simultaneous cytoplasm and nucleus membrane distribution of HNbTRAPα through the mouse antiserum against the central region (233–327 aa). C1~C6 were detected by the mouse antiserum against the C terminal (569–742 aa) of HNbTRAPα. D1~D6 were negative controls, which used negative antiserums as the primary antibody. The cytoplasm was labeled by the NbTubulin-*β* antiserum, and the cell nucleus was stained by DAPI. The white arrows indicate HNbTRAPα distributing in both the cytoplasm and perinuclear region of *N. bombycis*, as for the merged green and red fluorescent presented yellow. The white arrowheads indicate the perinuclear region distributed HNbTRAPα.

**Figure 5 jof-09-00407-f005:**
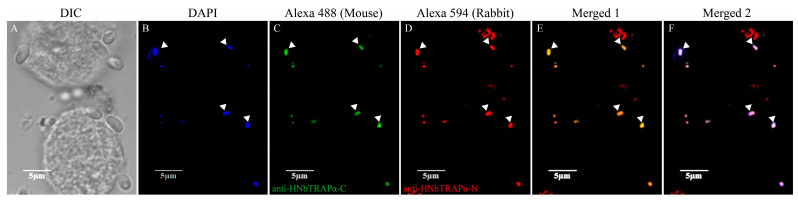
Indirect immunofluorescent assay of HNbTRAPα in mature spores. Pictures A~F were BmE cells infected by *N. bombycis* and many mature spores were released. HNbTRAPα-C mouse antiserum and HNbTRAPα-N rabbit antiserum were used as the primary antibody, respectively, and coupled with goat anti-mouse Alexa 488 or goat anti-rabbit, respectively. HNbTRAPα-C antiserum, as well as HNbTRAPα-N antiserum, recognized the nucleus region of the mature spores and their signals merged completely (white arrowhead).

**Table 1 jof-09-00407-t001:** List of primers used in this study.

Primer	Primer Sequence (5′-3′)	Amplicon Region or Initiation Site (bp)
*HNbTRAPα*-N_F	CGGGATCCATGACAAAAGCCCTTCATAA	1–696
*HNbTRAPα*-N_R	GCGTCGACAGGATTTCGAGCCATTAGATG	
*HNbTRAPα*-M_F	CGCGGATCCATGCCTTACAATTCATTC	697–981
*HNbTRAPα*-M_R	GTCGTCGACGTTTATTTTTGGCTGTAA	
*HNbTRAPα*-C_F	CGGGATCCATGGAGACTATTGAGAGC	1704–2226
*HNbTRAPα*-C_R	GTCGTCGACATTATACAAGCGTATTAT	
*HNbTRAPα*-GSP1	GGAGAAATGAAGCAGACACC	229
*HNbTRAPα*-GSP2	CAGATAGAAGCAGCAGTAGG	376
*HNbTRAPα*-GSP3	GGATGATGAAGAAGAGGAGG	405
*HNbTRAPα*-GSP4	AGAGAGTGCGTAATAGGGAG	443

## Data Availability

Not applicable.

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
