# Peer review of "A Putative TRAPα Protein of Microsporidia Nosema bombycis Exhibits Non-Canonical Alternative Polyadenylation in Transcripts"

_jof, 2023, doi:10.3390/jof9040407_

Round 1
Reviewer 1 Report
This paper is a paper that includes essential items about the Translocon of Microsporidia. It is worthy of posting, but it is necessary to renew some items.
1)Materials and Methods: If you use a kit, you don't need to describe the part in the manual. Describe the modified methods using the equipment so that you can see them.
2) Table 1: Display Primer and Primer Sequence to the left of the column.
3) Figure 1 ~ Figue4 is blurred. Please resubmit an image with a little more resolution.
I think that the content of the text is no problem.
Author Response
We appreciate you for spending time to review this article. You have given us professional advices and improved the quality of the article. We are very sorry for the spelling problem in the previous version, and those problem have been corrected in revised manuscript.
The following are our responcse to your comments.
Point 1: Materials and Methods: If you use a kit, you don't need to describe the part in the manual. Describe the modified methods using the equipment so that you can see them.
Response: Thanks for your suggestion. We have deleted the description that the same as the kit instruction.
Line 107-112: A portion of the midgut was ground and detect the infection of N. bombycis by microscopy. Then three severely infected tissues were ground in liquid nitrogen. Moderate powdered mixture was immediately transferred into RNase-free centrifuge tube, which was added CPL/ β-mercaptoethanol beforehand. The genomic DNA and total RNA of infected midgut were extracted by OMEGA E.Z.N.A.TM Plant DNA/RNA Kit (R6733-01) follow the instruction.
Point 2: Table 1: Display Primer and Primer Sequence to the left of the column.
Response: Thanks for your suggestion. The Primer and Primer Sequence were moved to the left of the column.
Point 3: Figure 1 ~ Figue4 is blurred. Please resubmit an image with a little more resolution.
Response: The Figures may have been compressed when inserted. They have been replaced by figures with appropriate resolution in revised version.

Reviewer 2 Report
In this study, the authors identified a gene in Microsporidia Nosema bombycis genome annotated as NbTRAPα which is a component of ER translocon machinery facilitating initiation of protein translocation. 3'-RACE analysis demonstrated the presence of two mRNA forms in microsporidia cells suggesting non-canonical alternative polyadenylation of transcripts after C951 or C1166 nucleotide respectively. This very interesting result as well as NbTRAPα localization on the nuclear membrane of microsporidia may be published in the journal after taking into account the following questions and remarks.
1. Among similar amino acid sequences identified in Genbank by BLAST analysis, only Encephalitozoon cuniculi homologous protein was also annotated as “signal sequence receptor alpha subunit”. The most part of homologous proteins of other microsporidia were characterized as uncharacterized, hypothetical proteins or thymidylate kinase (N. apis, E. bieneusi, N. ceranae). The most similar N. granulosis protein was annotated as “ovarian abundant message protein”.
Since NbTRAPα of all microsporidia species (1) do not have SP and TM domain, (2) are larger than homologous proteins of other organisms, (3) show a tendency to associate with the nuclei (the present study), it is necessary to provide additional evidence for the participation of this protein in the ER-translocation of polypeptides or to discuss its functional category more carefully.
2. The origin of the gene sequence used in this study for analysis, primer design and PCR must be indicated. Two full-length genes of this protein may be found in GenBank (Papilio xuthus sequencing project)
Papilio xuthus uncharacterized LOC106127269 (LOC106127269), mRNA, XM_013325354.1 52499-54727, (742 aa, 85.72 kDa, pI 8.52)
Papilio xuthus uncharacterized LOC106128892 (LOC106128892), mRNA, XM_013327357.1 314047-316272 (741 aa, 85.94 kDa, pI 8.61).
The properties of these proteins are slightly different from the values given in the article for the studied protein (743 aa (Fig 2), pI 8.78).
Probably, such gene and encoded protein may be obtained from Nb genome project contig KB908940.1 (100198-102428) by TA eliminating (as in P. Ñ…uthus variants) or adding some nucleotide in position 101730 between ORFs EOB14260.1 and EOB14261.1.
3. If the maximum protein size (1166/3) after non-canonical polyadenylation is 380-390 aa, as Abs to the fragment 569-742 aa can specifically recognize NbTRAPα in Sf9 cells (Fig.5 C)? Please, indicate the name of insect cell culture in Materials and Methods.
4. Since Abs work so well in immunolocalization, it would be very informative to estimate the mol. size of recognized proteins using a Western blotting. It would confirm the fact of non-canonical polyadenylation in microsporidia cells.
Author Response
We appreciate you for spending time to review this article. You have given us professional advices and improved the quality of the article. We sincerely thank you for that.
The following are our responcse to your comments.
Point 1: Among similar amino acid sequences identified in Genbank by BLAST analysis, only Encephalitozoon cuniculi homologous protein was also annotated as “signal sequence receptor alpha subunit”. The most part of homologous proteins of other microsporidia were characterized as uncharacterized, hypothetical proteins or thymidylate kinase (N. apis, E. bieneusi, N. ceranae). The most similar N. granulosis protein was annotated as “ovarian abundant message protein”.
Since NbTRAPα of all microsporidia species (1) do not have SP and TM domain, (2) are larger than homologous proteins of other organisms, (3) show a tendency to associate with the nuclei (the present study), it is necessary to provide additional evidence for the participation of this protein in the ER-translocation of polypeptides or to discuss its functional category more carefully.
Response 1: Thank you for your reminding and suggestions. Indeed, we should deliberate the protein more strictly and provide the relevant background for the identification of this gene. This gene is annotated TRAPα by BLAST and synteny analysis with genome data of Encephalitozoon cuniculi. Considering there are no TRAPα related domain information (might chaged because of database update), signal peptides and transmembrane domain containing in the gene, as well as functional research haven’t performed in the study, a hypothetical TRAPα protein of N. bombycis (HNbTRAPα) was used in the modified manuscript.
The “thymidylate kinase” annotation of N. apis, E. bieneusi and N. ceranae homologous proteins can not provide valuable information for protein function. Because the sequence coverage of proteins is very low and there are no related domain predicted using NCBI CD-Search. As for the similar protein in N. granulosis, it is annotated as “ovarian abundant message protein” might because it was identifed from the ovarian of infected host.
Point 2: The origin of the gene sequence used in this study for analysis, primer design and PCR must be indicated. Two full-length genes of this protein may be found in GenBank (Papilio xuthus sequencing project)
Papilio xuthus uncharacterized LOC106127269 (LOC106127269), mRNA, XM_013325354.1 52499-54727, (742 aa, 85.72 kDa, pI 8.52)
Papilio xuthus uncharacterized LOC106128892 (LOC106128892), mRNA, XM_013327357.1 314047-316272 (741 aa, 85.94 kDa, pI 8.61).
The properties of these proteins are slightly different from the values given in the article for the studied protein (743 aa (Fig 2), pI 8.78).
Probably, such gene and encoded protein may be obtained from Nb genome project contig KB908940.1 (100198-102428) by TA eliminating (as in P. Ñ…uthus variants) or adding some nucleotide in position 101730 between ORFs EOB14260.1 and EOB14261.1.
Response 2: Sorry for our careless that we did not provide enough information of the identification of the gene. Acturally, the protein EOB14260.1 annotated as signal sequence receptor alpha subunit (TRAPα as well) in database of NCBI is N-terminal of HNbTRAPα (1-232 aa) in this study. While EOB14261.1 is the other partial of HNbTRAPα (233-742). We found two different reads matching EOB14260.1 in genomic database of Nb. The longer one is consist of EOB14260.1 and EOB14261.1. Then the primers target to 3’ of EOB14260.1 and 5’ of EOB14261.1 were designed and used to amplify segments from gDNA and cDNA of N. b. As expected, the same sequences that cover the two predicted genes were amplified from both gDNA and cDNA (please see the fig. below). This is the origin that we suspect there are non-canonical polyadenylation happens in the transcription of this gene. We have submitted the sequence of HNbTRAPα recently (Genbank. OQ632563) and add the new access number in the revised version. The primers designed for this study were list in Table 1.
About the homologus proteins in Papilio xuthus, we know that genome of a microsporidium that is very similar to N. b contaminates genome of the butterfly. The two uncharacterized mRNA in Papilio xuthus should come from the microsporidium commensal with butterflies
Point 3: If the maximum protein size (1166/3) after non-canonical polyadenylation is 380-390 aa, as Abs to the fragment 569-742 aa can specifically recognize NbTRAPα in Sf9 cells (Fig.5 C)? Please, indicate the name of insect cell culture in Materials and Methods.
Response 3: As showen in Western blot results, the antiserum against peptides 569-742 (C terminal, HNbTRAPα-C) can mainly recoganize a ~37 kDa protein band and a weaken ~85 kDa in purified mature spores (Figure A lane 1). While in infected Sf9 cells containing N. b that are predominant in the proliferative phase, HNbTRAPα-C antiserum can specifically bind with a ~45 kDa protein (Figure A lane 2 and Figure B). The protein ~37 kDa, ~45 kDa and ~85 kDa are consistent with the predicted weight of the isoform that polyadenylated at 951 bp, 1167 bp and the full-length of HNbTRAPα.
Though we identified two types of non-canonical polyadenylation in our study, there are may not only two isoforms of HNbTRAPα in natural conditions. Because we have identified more than two isoforms of HNbTRAPα in different sources of N. bombycis materials. The two reported in the study are exsit in all the detected materials. This might the reason of other week bands can be reacted with antiserium.
The cultured insect cells are Sf9, and embryo cell BmE-SWU of silkworm Bombyx mori. The description had been add in Materials and Methods.
Western blotting assay of HNbTRAPα-C in N. bombycis mature spores and infected Sf9 cells that N. bombycis are predominantly in the proliferative phase.
Point 4: Since Abs work so well in immunolocalization, it would be very informative to estimate the mol. size of recognized proteins using a Western blotting. It would confirm the fact of non-canonical polyadenylation in microsporidia cells.
Response 4: Thanks for your suggestion. Western blot described in Response 3 had been add in the Materials and Methods, as well as Results.
